# Circular Economy in Home Textiles: Motivations of IKEA Consumers in Sweden

**Matthias Lehner \***, **Oksana Mont**, **Giulia Mariani** and **Luis Mundaca**

The International Institute for Industrial Environmental Economics (IIIEE), Lund University, P.O. Box 196, SE-221 00 Lund, Sweden; oksana.mont@iiiee.lu.se (O.M.); marianigiulia5@gmail.com (G.M.); luis.mundaca@iiiee.lu.se (L.M.)
* Correspondence: matthias.lehner@iiiee.lu.se

**Abstract:** (1) If policy-makers and businesses are to encourage consumers to participate in circular consumption systems, knowledge is needed about what motivates consumers to choose different disposal options. This paper aims to shed light on what motivates consumers to engage in circular home textile disposal practices. (2) Quantitative data was collected through a survey of members of the IKEA Family programme (N = 238), and logistic regressions were carried out to complement the quantitative analysis. Qualitative data was collected in semi-structured interviews with a total of 24 Swedish consumers. (3) Our findings show that donating and discarding are the most common options for handling home textiles, followed by reusing/storing, repairing, and reselling. Regression results indicate that environmental concerns, convenience, and economic reasons are the dominant motivations in choosing a disposal option. Prosocial behaviour and normative issues play a lesser role. (4) We recommend that policy-makers and businesses work to increase convenience of consumers' participation in circular product practices, and continually communicate environmental benefits of circular disposal options. Businesses and policy-makers aiming to make circular consumption more attractive also need to ensure economic benefits for consumers.

**Keywords:** circular economy; home textiles; circular waste management; circular textile practices; consumer motivation

## 1. Introduction

Consumption is the source of ever-increasing resource use and waste creation. The apparent inability to significantly reduce global consumption levels has led to attempts to find alternative ways to reduce impacts of consumption. The idea of a circular economic system that minimises resource input and waste, emissions, and energy leakage, by slowing, closing, and narrowing material and energy loops [1], has attracted attention and entered the policy discourse of different countries and supranational institutions. The Swedish Government, for example, first acknowledged the need for a circular economy-based societal development in a Government Bill from 1993 [2]. More recently, the European Commission (EC) updated its Circular Economy Package, in which a section titled 'Closing the loop—An EU action plan for the Circular Economy' [3] specifically addressed the incentivising of repair levels/durability and disassembly of products. Recycling targets were set at 60% by 2025 and 65% by 2030 (defined by the weight of the waste that enters the recycling/reuse system). In 2016, the EC also published an Eco-design Working Plan, starting from electronic products and later applying to other product groups [4].

Businesses have also started to recognise the circular economy as a potential solution that will enable them expand their operations while challenging the increasing consumption levels that result in unsustainable development. Suggested opportunities arising from the implementation of circular

business models include: increased productivity [5]; net material cost savings in the production phase [5]; increased price stability and security of supply [5]; improved brand image [6]; new market potential [6]; new demand for services [6]; increased interaction with consumers [6]; new job creation [5]; reduced consumption of raw materials [6]; reduced costs and risks from emissions, waste, and environmental legislation [6]; and the potential to attract new investors [6].

Given the relevance of textile consumption for sustainable development [7,8], we consider this area of research to be highly relevant. After the three big consumption areas (food/drinks, transport, and housing), textile consumption is estimated to have the highest environmental impact for the average European consumer [8]. According to the European Commission (EC), textile waste is a priority issue, and consumers are identified as key economic actors in driving the process from a linear to a circular economy. The EC particularly recognises that, through their purchasing, use and disposal choices, consumers can support or hinder the uptake of the circular economy. These choices are shaped by the information they have access to, the range and prices of existing products, and the regulatory framework [3]. In 2015, European consumers bought 6.4 million tons of new textiles (12.66 kg per capita) [8], and disposed of 5.8 million tons [3]. Most of these textiles are incinerated or discarded in landfills, with estimated collection rates for reuse or recycle no higher than 15–20% [8].

In Sweden, textile consumption is increasing steadily [9], and the climate impact of textile consumption increased by 27% between 2000 and 2017 [10]. The Swedish Environmental Protection Agency has reported that 90% of all climate impact is the result of purchasing new garments, and that a shift to more circular consumption patterns (using for longer, buying second-hand) could significantly reduce this impact. About a quarter of all textile consumption in Sweden comprises home textiles [11]. Textiles collected from donations in Sweden are usually accumulated and sorted in a central storage area and later distributed, partly at national level and partly abroad. Approximately 26,000 tons (3 kg per citizen) go to the second-hand market, handled mostly by Non-Governmental and Charity Organisations authorised by the Swedish municipalities. Major actors are *Myrorna*, *Erikshjälpen*, *Röda Korset*, *Läkarmissionen*, *PMU Intertrade*, *Stockholms stadsmission*, *Humana Sverige* and *Emmaus Björkå*. Around 70% of the textile collected by organisations (19,000 tons, 2.1 kg per citizen) is shipped from Sweden, mainly to Eastern Europe, Germany, the Netherlands and the Baltic States [12]. Only 3000 tons of textiles are redistributed to consumers by second-hand stores and charity shops in Sweden. The remaining 70,000 tons from users and 4,000 tons from the second-hand market end up in the waste management system and are incinerated [13]. The Swedish Environmental Protection Agency (EPA) has estimated that approximately 60% of incinerated textiles are in sufficiently good condition to be reused or recycled.

New business models must be developed, accepted, and supported by consumers if reuse and recycling are to be facilitated. For example, IKEA, the Swedish furniture retail giant, is one of the companies that has identified circular economy principles as a promising way to combine economic success with sustainability aspirations (https://highlights.ikea.com/2017/circular-economy/). Implementing circular economy principles in companies requires a good understanding of what motivates consumers when disposing of home textiles. Consumers need to change their behaviour, both as suppliers of raw material (returning used items) and as buyers of recycled, refurbished or used items.

Currently, little is known about motivations behind the disposal of textiles, especially home textiles. Laitala [14] conducted a literature review of disposal behaviour, and could not identify a single scientific paper examining practices and motivations for handling home textiles. This paper aims to fill this knowledge gap by focusing specifically on disposal behaviour and motivations of consumers in relation to home textiles. We describe the disposal practices of private consumers and identify factors that influence their disposal choices. The paper also aims to compare home textile-related behaviour to other, better-understood areas of textile disposal such as clothing. We use customers of IKEA Sweden as our sample to answer the following research questions:

- What are the home textile handling strategies of IKEA's customers in Sweden?

- What factors influence how IKEA's Swedish customers handle home textiles?

By answering these two questions, this paper can provide more general insights into this important but little studied field of textile consumption, and provide recommendations for policy-makers and business managers.

In the next section, a literature review of consumer textile disposal behaviour is presented, after which we explain our methodological approach. We then present the findings of our empirical work. Finally, we put those findings into the wider context of textile consumption, and conclude with recommendations for policy-makers and business managers.

## 2. Literature Review

While most literature on disposal behaviour has so far been focused on garments e.g., [14–25] and household waste e.g., [26–32], the subject of home textile disposal has received very little attention in academic research. Our literature study is therefore based on these two areas.

In one of the earliest papers on product disposal, Jacoby et al. [33] argue that disposal behaviour is influenced by certain factors—personal (attitudes, values, demographics), product (costs, value), and situational (storage space, infrastructure) factors. They suggest that, once they have decided to dispose of a textile, individuals have various options: (1) keep the textile stored until a suitable disposal option is available, or use it for other purposes; (2) get rid of it temporarily by lending it; and (3) get rid of it permanently by donating it, selling it, giving it away, or simply throwing it away with household waste.

In their study of disposal behaviour of young fashion consumers, Morgan and Birtwistle [24] identify another crucial factor that influences disposal behaviour: increasing purchase frequency. As fashion consumption has dramatically increased over the years, particularly since the advent of 'fast fashion', garment consumers are faced with an increased inflow of new items into their wardrobes, and thereby increased need to dispose of older items in their possession. Studying young fashion-sensitive consumers in South Korea, McNeill et al. [34] observe that these consumers—while generally displaying a strong willingness to quickly discard garments with little awareness of consequences—make an exception for items that are particularly important for their fashion identity, and go to length to repair these items.

Johnson, Young Lee, Halter, and Ju [20] suggest that, when consumers opt for disposal, they can compensate for their sense of guilt, e.g., due to environmental awareness, by storing items rather than disposing of them immediately. Textiles are then discarded only when a justifiable occasion occurs, such as an annual cleaning-out of cupboards and cabins, moving house, or seasonal change, i.e., all situations when discarding feels more justified. Similarly, Ha-Brookshire and Hodges [19] argue that whether or not environmental concern leads to behaviour change depends on whether the individual has a sense of guilt about their consumption. Morgan and Birtwistle [24] show that, where consumers lack awareness about textile waste, impact, and individual responsibility, a lack of sense of guilt can be observed. Even Joung [21] reaches a similar conclusion.

Morgan and Birtwistle [24] find that environmental awareness not only leads to a sense of guilt when disposing of textiles, it can also increase the likelihood of items being donated to charity, as it results in a positive feeling. Several other studies also argue that both environmental awareness, environmental attitudes, and individual values are relevant for disposal behaviour [26,30,35,36]. Other studies [19,23,37,38] document a general predisposition for consumers to avoid discarding clothing when they opt for more sustainable alternatives. König [39] identifies environmental motivations as an important factor explaining why consumers engage in repairing clothes. Supporting König's findings Scott and Weaver [40] show that consumers expect manufacturers to support repair efforts, while suspecting that companies make repair intentionally difficult.

Environmental awareness also appears to favour reselling of textiles. Cervellon, Carey and Harms [41] find that participation in second-hand markets is supported by concerns for the environment. Bianchi and Birtwistle [15] find that environmental awareness is a motivator for selling to second-hand

shops, but they also point out that only a few respondents in their study practiced reselling. In general, it appears that attitudes towards specific behaviour (e.g., recycling) are more effective predictors of behaviour than general positive attitudes towards the environment [15,31,42]. Gwozdz, Netter, Bjartmarz, and Reisch [43], for example, find that, despite young Swedes' high environmental concern and awareness, this knowledge does not translate into disposal habits.

Next to environmental motivations, individuals' disposal behaviour can also be influenced by prosocial motivations. Several scholars address disposal behaviour as a socially motivated behaviour, on account of perceived benefits to society. Chen et al. [44] report an increased effect of recycling information on behaviour of individuals with social motivations. Van Lange, Bekkers, Schuyt, and Vugt [45] find a similar result when they compare the inclination of individuals to donate (clothes, but also other valuables) to their social value orientation (prosocial, individualistic, and competitive). They find that prosocial individuals are more inclined to donate than other individuals. Mitchell, Montgomery, and Rauch [46] also find that donation is a disposal behaviour partly driven by a desire to help a fellow human being.

Another socially-influenced motivational force derives from extrinsically-derived social norms. A review by Ekström [47] concludes that consumers learning about circular textile practices through contact with role models at an early age are fundamental factors influencing sustainable behaviour as adults. Ekström adds that secondary socialisation (i.e., the process of learning appropriate behaviour as a member of a smaller group within the larger society) can be a determinant in stimulating sustainable consumption and disposal behaviour among adults. Andreoni [48] finds that individuals recycle and donate items for the sake of respect in their community, and according to Thøgersen [49] individuals act responsibly because of their beliefs about what is "the right or wrong thing to do" (p. 537).

Economic costs or benefits from different behavioural options are another important influential factor in disposal behaviour. Joung and Park-Poaps [22] find that reuse and resell of clothes among American college students is correlated to economic reasons. They explain this with the motivation that study participants wanted to save money through their textile handling behaviour. Goudeau [18] and Shim [25] find similar results for reselling garments. Many disposal options often associated with environmental awareness—such as reuse, repair, and reselling—can be related to economic motivations [50]. Diddi and Yan [51] find that American consumers perceive high costs associated with repairing clothing to be a major inhibitor for consumers repairing their clothes.

Household recycling literature illustrates how economic incentives influence the likelihood that individuals will recycle their waste or return bottles, cans or other products covered by a deposit scheme. Viscusi, Huber, and Bell [32] examine motivations for recycling plastic water bottles, by evaluating the importance of personal values, social norms, and economic incentives. They conclude that economic incentives for recycling have a greater influence on individuals' recycling behaviour than either private values or social norms. Plastic water bottle deposit laws have the greatest effect on recycling rates. Bor, Chien, and Hsu [27] examine the effects of a market-incentives recycling system for metals, paper, glass and plastic in Taiwan, and their results confirm its effectiveness. In a study of price incentives to encourage recycling among households in Portland, Oregon, Hong, Adams, and Love [28] also confirm the effectiveness of economic incentives in encouraging recycling.

Finally, disposal behaviour is also affected by situational variables, e.g., absence of infrastructure for waste sorting and collection, difficulty in identifying collection points, and lack of transport means to reach a collection point [16,17,19,23,25]. Some cities/neighbourhoods might not provide adequate services for waste sorting and collection, limiting the possibilities for the individual to turn their attitude and motivation into behaviour. Local legislation and information are other fundamental factors influencing and predicting textile handling behaviour [52].

Ha-Brookshire and Hodges [19] argue that necessity for storage space is a key factor when analysing consumers' decisions on textile handling options. This relates to the increase in consumption and the consequential increasing need to dispose of older textiles, combined with an unwillingness to discard them and therefore the choice to temporarily store textiles. In this scenario, storage space is an

obvious limiting factor to such (temporary) storage behaviour. Domina and Koch [16] also identify the lack of storage space as a reason to discard textiles.

Situational variables can also hinder intended pro-environmental behaviour. For instance, an individual who is committed to environmental causes may still be unable to sort waste and recycle, due to lack of time, storage space in the house or difficulty in reaching a designated recycling station. Studies by Shim [25], Koch and Domina [23], Domina and Koch [16,17], Ha-Brookshire and Hodges [19], and Morgan and Birtwistle [24] all conclude that convenience is one of the most influential factors driving consumers' behaviour.

Literature on garment disposal behaviour and waste recycling behaviour identifies a range of disposal strategies and motivational explanations. Consumers engage in reuse/storage, resell, donate, repair, and discarding behaviour. Behavioural motivations range from environmental concerns, to prosocial reasons, normative issues, economic reasons, and convenience/time/situational factors. However, literature does not explore connections between motivations and strategies, and does not explain which motivations and behaviours are most dominant for home textile disposal.

From a general perspective, it can be observed that all the factors or motivations (e.g., economic, environmental, normative issues) have the potential to play a horizontal or homogeneous explanatory role for each home textile behaviour (i.e., reuse, repair, donate, etc.). We hypothesise that this is *not* the case and that all identified motivations are not necessarily significant. Our alternative hypothesis is that a heterogeneous picture of home textile waste management behaviour appears if motivations are analysed in more detail. There is reason to expect, for example, that reuse/storage is influenced by economic reasons [22] as well as situational factors [16,19]. We expect repair of home textiles also to be primarily influenced by economic reasons [50]. We believe donation to be influenced by social sustainability reasons [45,46] and environmental concerns [24]. Reselling can be influenced by economic reasons [22] and environmental concerns [15,41]. Finally, discarding as a behavioural option could be expected to be influenced primarily by situational factors [16,17,19,25].

Our study aims to increase the resolution of motivational aspects driving home textile behaviour, and hypothesises that:

**Hypothesis 1 (H1).** *Reuse/storage of home textiles is influenced by economic reasons and convenience, time, and situational factors.*

**Hypothesis 2 (H2).** *Repair of home textiles is influenced by economic reasons.*

**Hypothesis 3 (H3).** *Donation of home textiles is influenced by prosocial behaviour and environmental concerns.*

**Hypothesis 4 (H4).** *Resell of home textiles is influenced by economic reasons and environmental concerns.*

**Hypothesis 5 (H5).** *Discard of home textiles is not only influenced by convenience, time, and situational factors, but also other motivational aspects (e.g., economic reasons).*

## 3. Materials and Methods

We followed a deductive approach in this research, starting from existing literature, to develop a conceptual understanding of the field and develop hypotheses. We decided on a mixed-method approach, applying both quantitative (a survey) and qualitative methods (interviews). According to Pole [53], mixed-method approaches provide the opportunity to study different aspects of the question while examining exploratory and confirmatory questions, thereby providing both depth and breadth in understanding a phenomenon. In our research, the mixed-method approach allows us to capture statistical explanations regarding behaviour and gain deeper understanding of these correlations, and uncover additional factors not emerging from the literature review.

For data collection, we were able to cooperate with IKEA as part of the pre-study phase of the *IKEA Textile Revival Project* (This project has ended. According to an IKEA employee who worked in this project, follow-up projects have since emerged relating to IKEA's ambition to create a circular business model), aimed at investigating consumer disposal behaviour regarding home textiles and identifying the most effective ways to involve consumers in circular systems. Data was collected with the involvement of IKEA, Transformator Design—a company that specialises in communication with consumers for the development of targeted services—and a master's student from Lund University (Giulia Mariani (2016). Influence of consumers' behavior on the circular economy application. The case study of a revival strategy for home textiles at IKEA. IIIEE Master's Thesis IMEN41 20162). Data collection focused on the company's domestic market in Sweden. Given its size and interest in developing its business towards more circular practices, IKEA provides a relevant case for the wider question of closing the material loop in consumption. The Swedish market is a suitable case for this study, as Swedish consumers are both aware and experienced in sorting and recycling different materials [54].

### 3.1. Survey

Quantitative data was collected through a survey (see Appendix A) targeting IKEA FAMILY members in Sweden. IKEA FAMILY is a consumer club with 2.7 million members in Sweden. The survey was sent out by e-mail link through the IKEA FAMILY club to 2105 individuals. A total of 404 responses were collected, but only 238 individuals completed the survey, resulting in a final response rate of 11.3%. Age representation was as follows: 18–25 years (3 respondents), 26–35 (47), 36–45 (56), 46–55 (62), 56–65 (44), and >65 (26). Of the 238 respondents, 150 were female, and 86 male, 161 had a post-secondary level education qualification, 71 had completed upper secondary school, and 6 had completed secondary school. Income ranged from > 40,000 SEK/month (31 respondents), to 20–40,000 SEK/month (127), < 20,000 SEK/month (42), to no personal income (4). Thirty-four respondents chose not to provide their income level. Sixty-seven of the respondents lived alone, 161 in a household with two adults, and 10 in households with three or more adults. Ninety-one of the respondents had children.

The survey aimed to understand customers' decision-making processes relating to underlying behavioural motivations. The survey asked respondents to specify how they had disposed of textiles in the past twelve months. Responses were given on a 5-point Likert scale (1—strongly disagree; 2—disagree; 3—neutral; 4—agree; and 5—strongly agree). A Likert scale is suitable for this research, as it permits collection and classification of large quantitative data on a scale. The approach is considered reliable, depending on the reported (significant) coefficients, by Shaw and Wright [55] and Robinson and Shaver [56], and is often used as a suitable tool for measuring attitude and intention [57–59].

The survey resulted in some initial findings that we both report and interpret in connection to our literature review and qualitative data. Subsections of our data lend themselves to complement our analysis using multinomial logistic regressions.

### 3.2. Logistic Regression

A multinomial logistic regression was used to support and complement the outcomes from the survey. The regression identified the motivational aspects that significantly enable us (or not) to predict circular home textile disposal practices. Based on the literature, and consistent with the survey, we examine home textile disposal behaviour and motivational aspects that can explain five different waste management options (i.e., dependent or outcome variables) available to textile users in Sweden: reuse/storage, repair, donate, resell and discard. All our dependent variables have more than two categories, so we used multinomial logistic regression procedures. We delimited our analysis to model significance, predictability, and likelihood statistics, to identify which motivations or factors significantly (or not) allow us to predict the outcome variables. Other aspects (e.g., individual parameter estimates, odd ratios) are outside the scope of this paper.

The predictors chosen for testing were 'economic reasons' (ER), 'prosocial behaviour' (PS), 'environmental concerns and awareness' (EC), 'convenience, time and situational factors' (CTS), and 'normative issues' (NI) (see Table 1).

**Table 1.** Independent variables and constructs with related items.

| Construct | Included item(s) |
|---|---|
| Economic reasons (*ER*) | $ER_1$: I often reuse/resell/repair/discard/donate home textiles for economic reasons |
| Prosocial behaviour (*PS*) | $PS_1$: It is very important for me to donate my home textiles to charity for people in need |
| Environmental concerns and awareness (*EC*) | $EC_1$: Textile manufacturing is responsible for the release of chemical pollutants in the water |
|  | $EC_2$: Air pollution can occur during some common dye processes of textiles |
|  | $EC_3$: The manufacturing process is highly water-intensive |
|  | $EC_4$: All kinds of textiles are recyclable |
|  | $EC_5$: Disposing of home textiles in a responsible way does not help with the reduction of raw materials use for new products |
|  | $EC_6$: I donate my home textiles to charity to do my part in decreasing the environmental problems |
|  | $EC_7$: I reuse home textiles because it can significantly benefit the environment |
|  | $EC_8$: To reduce environmental problems, I sell my unwanted home textile rather than throwing it away |
|  | $EC_9$: I try to repair my old home textiles because throwing away can significantly contribute to environmental problems |
| Convenience, time, and situational factors (*CTS*) | $CTS_1$: I reuse home textiles because it is not a hassle to me |
|  | $CTS_2$: It is not time-consuming to donate my home textiles to charity |
|  | $CTS_3$: I find it convenient to throw away unwanted home textiles |
|  | $CTS_4$: I am willing to spend time to resell, donate, and reuse my old home textiles |
|  | $CTS_5$: Reselling, donating, and reusing home textiles are not a trouble for me |
| Normative issues (*NI*) | $NI_1$: People important to me think that I should resell/donate/reuse home textiles |
|  | $NI_2$: People should be encouraged to resell, donate, and reuse home textiles |
|  | $NI_3$: More information about ways to resell, donate, and reuse home textiles should be made available by authorities to influence norms [1] |

[1] A reviewer observed that the wording of NI3 was insufficiently clear in the original questionnaire. The authors carefully assessed the issue and concluded that it would be best to add some additional text after the fact to clarify NI3, so '*by authorities to influence norms*' was added to NI3. These modifications were made to capture the intended purpose of NI3.

The Cronbach's alpha reliability statistics for all five constructs (i.e., independent variables) showed values in the range 0.65–0.73 (see Table 2). Ideally, Cronbach's alpha coefficient value should be above 0.7 [60], so our results are close to an acceptable internal consistency [61]. However, Pallant [62] observed that, for constructs with fewer than ten items (in our case $EC = 9$; $CT = 5$; $NI = 3$), relatively low Cronbach values ($\alpha < 0.5$) are common, and inter-item correlation values are needed to complement and provide a more appropriate indicator of reliability. In our case, inter-item correlation values were positive and in the range 0.24–0.48 (see Table 2). This is consistent with Briggs and Cheek [63], who showed that inter-item correlation optimal values should range from 0.20 to 0.40 for constructs with limited items.

**Table 2.** Reliability coefficient values ($N = 238$).

| Construct | Cronbach's Alpha | Interitem correlation |
|---|---|---|
| Environmental Concerns (*EC*) | 0.68 | 0.24–0.48 |
| Convenience, Time, and Situational factors (*CTS*) | 0.73 | 0.35–0.45 |
| Normative Issues (*NI*) | 0.65 | 0.48 |

For the 'Environmental Concerns' (*EC*) construct, the following items were kept for the regression analysis: $EC_6$, $EC_7$, $EC_8$ and $EC_9$. Under the 'Convenience, Time, and Situational factors' (*CTS*) construct, $CTS_1$, $CTS_2$, $CTS_4$ and $CT_5$ were retained. For 'Normative Issues' (*NI*), $NI_2$ and $NI_3$ were retained. Mean scores were 3.4 (SD = 0.93) for *EC*, 3.9 (SD = 0.86) for *CT*, and 4.5 (SD = 0.63) for *NI*.

Based on the above, our theoretical model for analysis is defined as follows:

$$\text{HTB}_{ij} = B_0 + X_{ER\,ij} + X_{PS\,ij} + X_{EC\,ij} + X_{CTS\,ij} + X_{NI\,ij} + \varepsilon, \tag{1}$$

where HTB is a dependent variable representing the home textile behaviour in each i disposal behaviour (i.e., reuse/storage, repair, donate, resell, or discard) by respondent j. $X_{ER}$ represents economic reasons, $X_{PS}$ prosocial behaviour, $X_{EC}$ environmental concerns, $X_{CTS}$ convenience, time, and situational factors, $X_{NI}$ normative issues, and $\varepsilon$ an error term. $B_0$ is a constant intercept. The entire survey used a 5-point Likert scale response choice (1= strongly disagree, 5= strongly agree) for all the analysed categorical variables, both dependent and independent (see Appendix A). To ascertain which independent variables would enable us to predict each HTB, likelihood ratios and chi-square statistics were calculated. Pseudo $R^2$ and overall accuracy (i.e., how well each model is capable of predicting outcome variable) were also estimated.

### 3.3. Interviews

A total of 24 interviews were conducted, aimed at obtaining a broader understanding of the perceptions and factors that influence the studied behaviour. The interview questions were broad, aimed at casting a wide net around shopping and disposal behaviour of home textiles. As with the survey data collection, we collaborated with IKEA regarding the interviews, and worked with the consultancy firm Transformator Design to develop the questionnaire. Interviews were conducted by Transformator Design in IKEA stores in Malmö (16) and Stockholm (8). The interviews followed a semi-structured questionnaire guideline (see Appendix B) but differed somewhat in the to-be-guaranteed level of detail. Interviews in Malmö lasted about 20–30 min, and the interviews in Stockholm lasted 60–90 min. The interviews in Malmö were spontaneous, while the interviews in Stockholm were pre-arranged with customers who had volunteered to participate in the study. Interview data was anonymised by Transformator Design/IKEA before we were granted access to it, to ensure confidentiality.

Interview Analysis

The interview transcripts obtained from IKEA were read and categorised according to the analytical framework to identify the main elements that influence and motivate disposal behaviour. Our coding aimed to identify common patterns among interviewees. Major patterns identified were 'moral obligation', 'knowledge of textile waste produced and its impact', 'convenience and habit', 'responsibility', and 'senses of pride and guilt'.

## 4. Results

This section starts with a presentation of most significant descriptive statistical results from the survey, followed by the regression results, and finally, the qualitative interview results.

### 4.1. Findings from the Survey

First, we wanted to find out the proportions of different disposal behaviours among the study participants. Participants were asked to report which disposal behaviours they had engaged in during the previous 12 months. Respondents could select more than one textile waste handling method. 'Donate' was the most common textile handling option, reported by 76.89% of respondents. The second most preferred option, reported by 55.46% of respondents, was to 'discard' home textiles. The other options, in order, were 'reuse/store' (51.26%), 'repair' (35.29%), and 'resell' (24.37%) (see Figure 1).

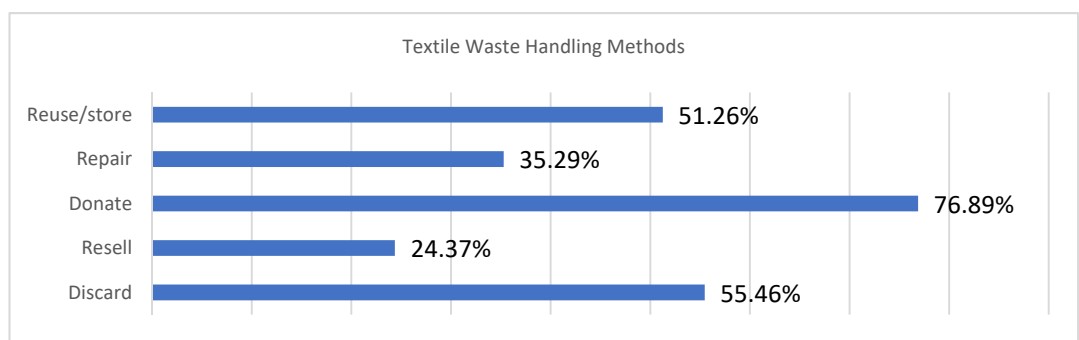

**Figure 1.** Textile waste handling methods.

We were also interested in assessing the level of environmental awareness among survey respondents. We found our survey population to be largely aware of environmental consequences of home textile waste, with 90% of respondents agreeing or strongly agreeing that textile manufacturing is responsible for chemical pollution of waterways, 80% (strongly) agreeing that it is a highly water-intensive process, and 74% (strongly) agreeing that air pollution can occur during dyeing processes. Our study participants did not necessarily perceive circular waste handling practices as a solution to these environmental challenges, with 58% (strongly) agreeing to the claim that even responsible disposal of home textiles does not help to reduce raw material use.

Despite this scepticism towards circular solutions, respondents were very much in favour of such practices: 97% (strongly) agreed that reselling, donating, and reusing home textiles were good ideas (82% also disagreed that these practices were more trouble than they are worth), and 95% agreed that people should be encouraged to resell, donate, or reuse home textiles.

In the questions about disposal motivations, respondents reported that they donate primarily for environmental reasons and to help people in need. They discard for convenience (and refrain from discarding for environmental reasons). They reuse/store for environmental and economic reasons, but they choose not to reuse/store because it is a hassle for them. They repair for environmental reasons, but some of them do not know how. Finally, respondents resell for environmental reasons, and (to a much lower degree) for economic reasons (see Figure 2).

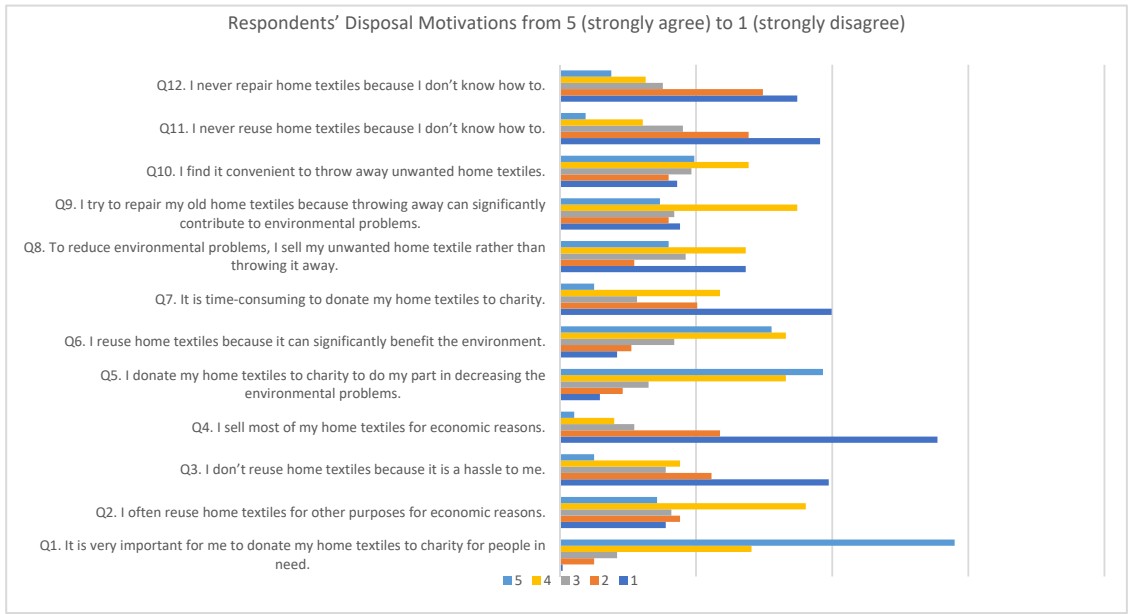

**Figure 2.** Respondents' disposal motivations.

We were interested in exploring whether the condition of the home textiles made any difference to respondents (i.e., reasons for disposing of them) (see Table 3). A clear distinction could be identified in the likelihood of discarding home textiles regardless of their condition. Respondents reported discarding home textiles in only 5% (good condition but want to change) or 6% (too many of something, e.g., towels) of cases, but 39% and 36% respectively would get rid of textiles if they had a permanent stain or were visibly used (i.e., faded colours).

**Table 3.** Preferred choices of disposal options for different scenarios.

| Disposal Situations | "Your Bed Linen A Hole." | "The Curtains in the Living Room Are in Good Conditions but You Want to Change Them." | "The Tablecloth Has a Stain That Doesn't Go Away." | "The Colour of the Chair Pads in the Kitchen is Faded." | "There Are Towels Taking Space in the Cupboard That Have Never Being Used." | "The Furniture in the Bedroom Has Been Changed and You Need to Get Rid of the Old Pillow Covers and Blankets." |
|---|---|---|---|---|---|---|
| Reuse/store | 56.72% | 29.41% | 52.94% | 35.71% | 35.71% | 31.93% |
| Repair | 22.69% | 0.42% | 2.52% | 10.92% | 0.42% | 0% |
| Donate | 7.14% | 64.70% | 18.07% | 29.41% | 61.34% | 66.39% |
| Resell | 0% | 21% | 2.52% | 4.20% | 15.13% | 19.33% |
| Discard | 30.67% | 5.04% | 38.65% | 36.13% | 5.88% | 11.34% |

Donations were most commonly reported for textiles that were in good condition but needed to be changed (65%), because the respondent had too many of something (61%), or had purchased a new textile to replace an old one (66%). For lightly-damaged goods (e.g., a hole in bed linen) 57% reported reusing the textile, and even for textiles with a permanent stain, 53% could imagine reusing them. Reselling and repairing were the least-reported options; reselling mostly concerned textiles in good condition that respondents wanted to change (21%) after an old item had been replaced by a newer identical one (19%); repairing was only reported for lightly-damaged goods (23%).

*4.2. Regression Results*

We then wanted to identify which motivations (i.e., predictors) significantly (or not) enable us to ascertain each behaviour option for home textile waste management. Results are summarised in Table 4. Each model contains all the five potential predictors identified in the literature.

**Table 4.** Significance of predictors to each model. Likelihood ratios and respective model accuracy and pseudo $R^2$ for each waste management behaviour. Chi-square statistics in parenthesis. *** $p < 0.01$, ** $p < 0.05$, * $p < 0.1$.

| Outcome Variable $Y_i$ | −2 Log Likelihood | | | | | | Overall Model Accuracy | Pseudo $R^2$ | |
|---|---|---|---|---|---|---|---|---|---|
| | *Model* | $X_{ER}$ | $X_{PS}$ | $X_{EC}$ | $X_{CTS}$ | $X_{NI}$ | | Cox and Snell | Nagelkerke |
| Reuse/store | 423.557 | 455.786 (32.22) *** | 427.529 (3.97) | 432.860 (9.30) * | 442.774 (19.21) *** | 426.293 (2.73) | 51.7% | 0.32 | 0.34 |
| Repair | 474.978 | 486.926 (11.94) ** | 478.560 (3.58) | 504.297 (29.31) *** | 486.781 (11.80) ** | 481.015 (6.03) | 42.9% | 0.32 | 0.33 |
| Donate | 280.747 | 283.575 (2.82) | 327.360 (46.61) *** | 290.803 (10.05) ** | 296.879 (16.13) *** | 294.969 (14.22) *** | 72.7% | 0.44 | 0.51 |
| Resell | 525.198 | 535.726 (10.52) ** | 533.484 (8.28) * | 535.332 (10.13) ** | 531.519 (6.32) | 535.316 (10.11) ** | 34.9% | 0.23 | 0.24 |
| Discard | 459.747 | 468.064 (8.31) * | 468.517 (8.77) * | 479.296 (19.54) *** | 476.016 (16.26) *** | 463.274 (3.52) | 47.1% | 0.33 | 0.35 |

All tested models are significant and able to predict the correct category for each type of home textile handling behaviour (i.e., reuse/store, repair, donate, resell, and discard) ranging from approximately 34% to 72%. For reuse/store behaviour, the model was statistically significant, $\chi^2$ (20, 238) = 91.96,

$p < 0.001$, where economic reasons ($X_{ER}$), environmental concerns ($X_{EC}$) and convenience, time, and situational factors ($X_{CTS}$) are significant predictors. To some extent, this confirms our hypothesis (H1) but shows that more attention should be paid to environmental concerns. The model explains between 32% (Cox and Snell R square) and 34% (Nagelkerke R square) of the variance of reuse, and can correctly classify 51.7% of the categories. Prosocial behaviour ($X_{PS}$) and normative issues ($X_{NI}$) do not make a unique, statistically significant contribution to the first model. The same applies for repair. The model is significant, $\chi^2$ (20, 238) = 91.69, $p < 0.001$, and it can be observed that the same predictors ($X_{ER}$, $X_{EC}$ and $X_{CTS}$) are significant. Our hypothesis (H2) is therefore only partially supported.

For donating behaviour, estimates show that all predictors, except economic reasons, are statistically significant. This confirms that people who are motivated to donate home textiles are not driven by economic motives or rewards; instead, prosocial behaviour ($X_{PS}$) is the strongest predictor. In this specific model, our hypothesis (H3) is partially supported, and results suggest that much more consideration should be given to other motivations. The donate model ($\chi^2$ [20, 238] = 141.46, $p < 0.001$) also exhibits the highest predictability level (72.7%) across all tested behaviour models.

Relatively speaking, and compared to the other tested models, the categories in the resell behaviour model ($\chi^2$ [20, 238] = 63.46, $p < 0.001$) can only be determined to a lesser extent (34.9%) by all predictors except $X_{CTS}$. This shows that consumers who resell home textiles are not motivated by convenience. It also suggests limited experience or preferences regarding the reselling of home textiles, confirming it as the least-preferred option for circular textile practices (see Figure 1). Our hypothesis (H4) is only partially confirmed.

Finally, the discard behaviour model ($\chi^2$ [20, 238] = 97.71, $p < 0.001$) correctly classified up to 47.1% of the categories (1: strongly disagree; 5: strongly agree) and explained up to 35% of the variance for this specific waste management behaviour. Whereas most predictors make a statistically significant contribution to the model, normative issues ($X_{NI}$) do not. This suggests, for instance, that home textile users that engage with this behaviour may do so irrespective of the prevailing social norm (e.g., avoid discarding of textiles). $X_{CTS}$ is only one of the four significant predictors, so our findings support the hypothesis (H5) that discard is influenced by multiple motivations, not necessarily convenience.

All tested models confirm our alternative hypothesis to various extents. There is heterogeneity across all the identified factors that motivate or influence consumers to handle home textiles. A vertical reading of the results in Table 4 indicates that environmental concerns ($X_{EC}$) and convenience, time, and situational factors ($X_{CT}$) tend to play a relatively more prominent role, followed by economic reasons ($X_{ER}$).

*4.3. Interview Findings*

The interviews were conducted to understand the broader context within which disposal behaviour takes place.

A first observation is that we identified a general lack of awareness of the problem regarding home textile waste in the interview data. Interviewees were convinced that they did not throw away many home textiles, so the act of disposal was not significant for them. They felt that home textiles, perceived as a rarely purchased and disposed item, merited less attention than everyday items like household waste. Swedes recognise household waste as being an important issue, and recycling rates for household waste are high [64,65]. Interviewees paid little attention to the disposal of home textiles, and disposal behaviour appears to be secondary to the stronger motivation for purchasing new products. As the need to dispose of old products is the consequence of a previous action to buy something new, the decision of how to dispose of them is triggered by another behaviour that the individual feels is more important. Interviews showed no concerns about disposal at the time of purchase. One respondent (male, aged 27) reported purchasing home textiles based on the desire to change colours and materials in the home. Disposal was no part of this decision, and the need to dispose of old items seemed to be an afterthought.

Another important aspect arising from the interviews is the confusion of individuals on what could be done with textiles they no longer wanted. Citing a young girl aged 19: "We throw away textiles. What should we do with them?" Only few interviewees reported basic knowledge about textile production and its consequent impact on the environment. Other interviewees described how they knew how to dispose of clothes by selling at flea markets, but that the same did not apply to home textiles: "We're also open to the idea of taking textiles to flea markets, but most often you sell clothes there and not towels and linen." (women, 40 and 60), and "We want to get rid of textiles that we no longer use. Used fabrics that are not clothes normally go in the rubbish. It's hard to give linen and towels and those things to charity, there's holes in them and all that." (women, 25 and 30). Several interviewees left decision-making on textile handling to another member of the family (often the wife or the mother). Those consumers who did report greater knowledge about textile production also reported a more proactive approach to disposal behaviour. This began already at the time of purchase when they claimed to consider social issues, fair trade, and product quality. Some also reported experience of textile return schemes and second-hand products.

While some interviewees felt justified in disposing of home textiles, citing a lack of time to deal with them in other ways, others who admitted they had no idea what to do with their unwanted textiles refused to throw them away, since they felt that it was not the right thing to do. Interviewees stated that they only disposed of the item after a period of storage if they could not find another use for the item or find someone to pass them on to. Some interviewees reported the practice of creating a personalised waste management hierarchy. Before getting rid of home textiles, they first find alternative ways to prolong their lives (e.g., use them as rags, move them to holiday homes, lend them to family and friends, and store them until they find a proper way to dispose of them). However, most interviewees seemed to put little effort into trying to find another disposal option, and only acted upon an external stimulus to choose a better option when an opportunity arose.

The interviews showed that this was connected with a feeling that disposing of home textiles in more sustainable ways was necessary but cumbersome, e.g., because of collection centres being too far away. Donating textiles felt like a suitable solution to this dilemma: "We have too many things. It feels so good to give away and know that it will be used for something that's good. Maybe for the benefit of someone else." (women, 25 and 30). Several of our interviewees expressed major concerns about knowing who benefits financially from donations. They trust charity organisations perceived to have a good reputation. Some interviewees would prefer municipalities to manage unwanted textiles rather than private companies.

At the same time, home textiles, in general, were an area of consumption with low levels of personal involvement, so consumers had little interest in dealing with the matter of disposal. Several interviewees indicated that, if a textile product has a low price or is perceived as having low value, then they felt justified in disposing of it more readily. Other reported factors influencing consumers' perception of value were brand, material, design, and heritage. One major finding of the interviews is that, when consumers do not perceive any value in the product, they think that the textile cannot be reused or donated, so they throw it away. Several interviewees did not donate, repair, or resell their worn-out textiles because they saw no functional value in them.

The interviews also showed that consumers select their textile handling options based on the type of product. For instance, curtains appeared to be of higher value, as they fill a functional as well as an emotional value in the home of the interviewed consumers. More time and money are invested in their purchase. When this textile is replaced due to fashion and aesthetic taste changes, they retain their functional value in the eyes of the consumer, and this helps to explain why respondents described donation as a common disposal strategy. In contrast, interviewees identify towels and blankets as products that are basic necessities in every house due to their exclusively functional value (except for cases in which highly valuable materials are used for these products). This kind of product usually has less emotional value attached, and when the time comes to dispose of these products, they are considered as retaining little value.

## 5. Discussion

Reflecting on the five disposal behaviours studied in relation to the circular economy, it is encouraging that 'donating' is the most favoured disposal option. We found that the preference for donating is primarily prompted by prosocial motivations. This is very much in line with findings of Nordlund and Garvill [65], Van Lange et al. [45], and Mitchell et al. [46], which all identify prosocial values as an important predictor for individuals' willingness to donate clothes and other valuables. The perception of a charity appears to be important—individuals donating their textiles cared about who would benefit from their act of perceived kindness. This raises questions about whether IKEA would be able to encourage donating within its operations, which are well-known by consumers to be for-profit, as opposed to the non-profit operations of a third party. Donating was the only behaviour in our study that did not show economic incentives to be a significant motivator for the behaviour.

Reuse/storage was the third-most reported behaviour among study participants. Our data suggests that this behaviour is supported by several of the motivational factors studied in this paper, including economic reasons, environmental concerns, and convenience/time/situational factors. Our interviews suggest that reuse/storage is not by definition a circular textile handling behaviour. In many instances, reuse/storage only serves as a temporary solution to deal with conflicting feelings of wanting to get rid of a textile while having a bad conscience about disposing of it. This can result in a stepwise process of disposal in which reuse and storage are one stage towards getting rid of an item when an acceptable permanent solution arises. When new products are purchased, old home textiles become redundant and are then temporarily stored in the home. During the storage period, the owners remain open to reusing them in another context or passing them on to a relative or friend. Jacoby et al. [33] support the conclusion that one common strategy for dealing with used textiles is to temporarily store them until a decision is made on how to dispose of them.

Our data also suggests that the type of disposal option ultimately chosen is often a result of what is convenient at the time when storage space is needed. Depending on the circumstances, reuse/storage in some instances only prolongs the time until discarding, rather than leading to a more circular waste handling method. This finding is in line with other studies (e.g., Ha-Brookshire and Hodges [19] regarding apparel). This suggests the importance of availability and accessibility of circular waste handling options in textile waste handling, something that is well documented for kerbside recycling. Research that has documented the significance of convenience in household waste recycling [30] also seems to be relevant to home textile disposal behaviour.

Less common disposal options were repairing and reselling. One possible explanation is that consumers are unfamiliar with these options. In interviews, we found evidence for limited familiarity with repair and resell as options when dealing with home textiles. Unlike clothes, home textiles are not perceived as an item that can easily be sold second-hand, and repair is inhibited by the often-perceived low value of home textiles compared to clothing. Johnson et al. [20] identify emotional attachment to the textile product as a factor strongly influencing the choice of disposal method [34], and our research suggests that home textiles carry low emotional attachment in general, even though there appear to be some sub-categories—such as curtains—that can be perceived as high-value at the moment of disposal.

Discarding home textiles—representing the least circular of textile waste handling options studied—was the second-most practiced behaviour among study participants. People who are most likely to discard home textiles are those least affected by social norms. This can be connected to an observed lack of awareness about the environmental consequences of home textiles, as well as a lack of knowledge about other behavioural options than discarding. It can be speculated that home textiles are not perceived as being exposed to social norm scrutiny in the same way as clothes or other possessions.

### Managerial and Policy Implications

From a policy perspective, interventions targeting convenience (such as education about disposal options, or easy access to circular disposal options) are necessary to encourage the choice of one disposal option over another, as convenience is a crucial factor in predicting participation in circular

waste management practices. Our research clearly shows that individuals need to know what to do with home textiles, and such behaviour must be reasonably time-efficient and convenient. This applies particularly for home textiles, which in many cases are items of low personal involvement. More convenient ways to manage home textiles sustainably are effective nudges for people to participate in circular schemes.

Policy-makers should (a) ensure that consumers are aware of the various disposal options, (b) make it easy for consumers to use the most environmentally-friendly options, and (c) make it inconvenient to dispose of textiles in an unsustainable way. One natural everyday solution would be more accessible recycling and donation points, as well as a functional infrastructure for repairing or reselling home textiles. Many Swedish municipalities already offer fashion waste collection next to household recycling stations. Such collection points could be visibly upgraded to include home textiles as well. Sweden has also lowered VAT on repair services, a policy that should be considered in other countries, as it increases the likelihood that businesses could be competitive in this space.

Information about the environmental consequences of home textile consumption is also important, as our results show that environmental motivations are among the most significant. At the same time, our interviews suggest a lack of awareness about problems associated with home textiles. Both businesses and policy-makers trying to encourage participation in circular disposal systems should continuously inform and remind consumers about the environmental significance of circular home textile disposal systems. For businesses, our research suggests that uptake is best where economic incentives are combined with environmental information, ensuring that consumers can trust that something good comes out of their behaviour. Some fashion retailers, for example, have already started to experiment with give-back schemes that take care of the textile. H&M has trialled such a scheme, where give-back is rewarded with vouchers. This could be adopted by home textile retailers as well, making more sustainable disposal behaviour both more environmentally beneficial and financially profitable.

While our research shows that, currently, it is not common for consumers to resell their used home textiles, this could change in the future with more innovative technological services that allow easy selling of home textiles. Technological innovation resulting in online platforms for reselling of unwanted textiles have been successful in several other textile-related markets (Online marketplaces such as 'Shpock' or 'Facebook Marketplace' allow for easy reselling of unwanted textiles. Many countries also have local online marketplaces, such as 'Blocket' in Sweden, or 'Gumtree' in the UK), though not so much regarding home textiles yet. However, the assumption that easy-to-use online reselling platforms will become more common even for home textiles appears reasonable. Businesses such as IKEA should actively engage with such technological solutions to encourage reselling.

Our research confirms that there is often a window of opportunity (i.e., storage) in which consumers can be reached with suggestions for other handling options than discarding. At any point during this window of opportunity, the tentative decision to discard the textile can be replaced with a more attractive option for the consumer, as long as the right motivational triggers (economic, convenience, environmental) are addressed. During this period, most consumers are open to considering other disposal options such as donation [19], thereby relieving their bad conscience. This is particularly true in cases where home textiles retain a perceived value.

Important in this respect is that we observed another component of decision-making regarding the disposal of home textiles. In households, there is often one person in charge of decisions regarding the disposal of home textiles, while other household members either know nothing or are not particularly aware or concerned about this process. This decision-maker is often identified as the "woman of the house" [29,43,66], and in our study too, some interviewees refer to "their mother" for this decision. Individuals concerned with disposal behaviour often create their own waste hierarchy, involving ranking options according to what is preferable and possible. Objects move up or down the waste hierarchy depending on the conditions under which a disposal decision must be taken. Encouraging more sustainable disposal behaviour of home textiles therefore involves reaching the right person in a

household. Businesses such as IKEA have a privileged position in being able to reach these individuals and engage with them, because they are the very same individuals that are their customers.

## 6. Conclusions

This study demonstrates that there is great scope to encourage and support consumers' participation in circular textile practices. Donating is already prominent and much used, and this fact should be used to further simplify and incentivise this option of circular home textile handling. We expect reselling to become a more prominent option, as information and communication technologies reduce the transaction costs for consumers to engage in such behaviour. Even repair could become more common as an option if this became more convenient and economically attractive. Our research also confirms the challenge in diverting behaviour away from discarding, which is the least preferred waste management practice from a sustainability perspective, but often the most convenient.

In terms of motivational factors, our research shows that environmental concerns, convenience, and economic reasons are most important factors when choosing how to dispose of home textiles, while prosocial behaviour and normative issues are less relevant.

We therefore conclude that more research is needed about motivational factors in disposal behaviour of home textiles. Our research gives reason to believe that towels, bed linen, pillows, curtains, or tablecloths are distinct in their value assessment by consumers and result in different cognitive and motivational setups regarding their disposal. We encourage other scholars to study disposal behaviour regarding home textiles, focusing on sub-categories to improve understanding of motivational factors and disposal options for these various sub-categories.

**Author Contributions:** Conceptualisation, M.L., O.M. and G.M.; Data curation, L.M.; Formal analysis, G.M. and L.M.; Funding acquisition, O.M. and L.M.; Investigation, G.M.; Methodology, M.L., O.M., G.M. and L.M.; Project administration, O.M.; Resources, O.M.; Supervision, O.M.; Visualisation, M.L.; Writing—original draft, M.L.; Writing—review & editing, M.L. All authors have read and agreed to the published version of the manuscript.

**Funding:** The analysis of data and writing of this article were funded by MISTRA Sustainable Consumption—from Niche to Mainstream (programme period 2017–2021). (Data was collected as part of a master's thesis and not funded by the programme.)

**Acknowledgments:** We want to thank the case company IKEA for their invaluable time and support for this study. We would also like to thank MISTRA Sustainable Consumption for financial support in conducting the research and regarding the open access fees. Luis Mundaca gratefully acknowledges funding from the Swedish Innovation Agency (grant 2018-04649).

**Conflicts of Interest:** The authors declare no conflict of interest. The funders and the case company (IKEA) provided resources for this study (i.e., access to customers and the free service of a consultancy they work with), so they indirectly participated in the collection of data. However, they had no role in the design of the study, in the analysis or interpretation of data, in the writing of the manuscript, or in the decision to publish the results.

## Appendix A

IKEA Family Members Questionnaire
The online survey was conducted in Swedish.
*Greetings!*

*You are receiving this message because you have been selected to participate in a brief survey on home textiles disposal habits. The responses are anonymous and will be used to support a research project at the International Institute for Industrial Environmental Economics at Lund University. The objective of the survey is to collect basic information on people's disposal habits with home textiles.*

*Below, and on the indicated link, you will find a short survey that will take approximately 10 minutes to answer. All respondents will be later selected for a raffle and will get a chance to win two discount vouchers of SEK 200 each to be spent in any IKEA store.*

*Thank you very much for your time and contribution!*

*For the purpose of this survey, here's a list of home textiles to keep in mind.*

- Rugs.

- Bedroom textiles: bed linen; comforters; bedspreads; blankets and throws; pillows; mattress and pillow protectors; canopies and bed tents; sleeping bags for babies.
- Curtains and blinds.
- Fabrics.
- Cushions and cushion covers.
- Kitchen textiles: kitchen towels; aprons; pot holders and oven mitts.
- Table linen: place mats; coasters; tablecloths and runners; chair pads.
- Bathroom textiles: towels; bath mats; shower curtains.
- Items such as carpets and textiles which are integrated parts of other products, such as furniture, are not included in this questionnaire.

In the last 12 months, have you used <u>at least one</u> of the following five textile handling methods? (thick boxes will be set next to each option. More than one option can be selected).

1. *Resell refers to selling textile items directly to other people, through consignment shops, to resale or second-hand shops, through online websites, and at garage sales or flea markets.*
2. *Donate refers to giving away textiles to family or friends. Donating can also be done through charitable organizations, thrift stores, curbside recycling programs, retail recycling programs, online companies.*
3. *Reuse/store refers to using textiles for a purpose other than for which it was originally intended. For example, old sheets may be used as cleaning rags around the house.*
4. *Repair refers to the act of fixing the textile either by yourself or by a professional.*
5. *Discard refers to when textile is thrown away, abandoned, or destroyed.*

Please select your level of agreement for each one of the following statements.

*Appendix A.1. Section 1*

**Table A1.** Environmental Impact of Textile.

| | Question | Strongly disagree (1)—Disagree (2)—Neutral (3)—Agree (4)—Strongly Agree (5) | | | | |
|---|---|---|---|---|---|---|
| 1 | Textile manufacturing is responsible for the release of chemical pollutants in the water. | 1 | 2 | 3 | 4 | 5 |
| 2 | Air pollution can occur during some common dye processes of textiles. | 1 | 2 | 3 | 4 | 5 |
| 3 | The manufacturing process is highly water-intensive. | 1 | 2 | 3 | 4 | 5 |
| 4 | All kinds of textiles are recyclable. | 1 | 2 | 3 | 4 | 5 |
| 5 | Disposing of home textiles in a responsible way does not help with the reduction of raw materials use for new products. | 1 | 2 | 3 | 4 | 5 |

*Appendix A.2. Section 2*

**Table A2.** Home Textile Disposal Motivation.

| | Question | Strongly disagree (1)—Disagree (2)—Neutral (3)—Agree (4)—Strongly Agree (5) | | | | |
|---|---|---|---|---|---|---|
| 1 | It is very important for me to donate my home textiles to charity for people in need. | 1 | 2 | 3 | 4 | 5 |
| 2 | I often reuse home textiles for other purposes for economic reasons. | 1 | 2 | 3 | 4 | 5 |
| 3 | I don't reuse home textiles because it is a hassle to me. | 1 | 2 | 3 | 4 | 5 |
| 4 | I sell most of my home textiles for economic reasons. | 1 | 2 | 3 | 4 | 5 |
| 5 | I donate my home textiles to charity to do my part in decreasing the environmental problems. | 1 | 2 | 3 | 4 | 5 |
| 6 | I reuse home textiles because it can significantly benefit the environment. | 1 | 2 | 3 | 4 | 5 |
| 7 | It is time-consuming to donate my home textiles to charity. | 1 | 2 | 3 | 4 | 5 |
| 8 | To reduce environmental problems, I sell my unwanted home textile rather than throwing it away. | 1 | 2 | 3 | 4 | 5 |
| 9 | I try to repair my old home textiles because throwing away can significantly contribute to environmental problems. | 1 | 2 | 3 | 4 | 5 |
| 10 | I find it convenient to throw away unwanted home textiles. | 1 | 2 | 3 | 4 | 5 |
| 11 | I never reuse home textiles because I don't know how to. | 1 | 2 | 3 | 4 | 5 |
| 12 | I never repair home textiles because I don't know how to. | 1 | 2 | 3 | 4 | 5 |

*Appendix A.3. Section 3*

**Table A3.** Home Textile Disposal Attitude.

| | Question | Strongly disagree (1)—Disagree (2)—Neutral (3)—Agree (4)—Strongly Agree (5) | | | | |
|---|---|---|---|---|---|---|
| 1 | Reselling, donating, and reusing home textiles are good ideas. | 1 | 2 | 3 | 4 | 5 |
| 2 | I am willing to spend time to resell, donate, and reuse my old home textiles. | 1 | 2 | 3 | 4 | 5 |
| 3 | More information about ways to resell, donate, and reuse home textiles should be made available by authorities to influence norms. [1] | 1 | 2 | 3 | 4 | 5 |
| 4 | Reselling, donating, and reusing home textiles are more trouble than they are worth. | 1 | 2 | 3 | 4 | 5 |
| 5 | People should be encouraged to resell, donate, and reuse home textiles. | 1 | 2 | 3 | 4 | 5 |

[1] A reviewer noted that the wording in NI3 was insufficiently clear in the original questionnaire. The authors carefully considered the issue and concluded that it would be best to add some additional text, so '*by authorities to influence norms*' was added to NI3. These modifications were made to capture the intended purpose of NI3.

*Appendix A.4. Section 4*

**Table A4.** About Home Textile Disposal.

| Question | Strongly disagree (1)—Disagree (2)—Neutral (3)—Agree (4)—Strongly Agree (5) | | | |
|---|---|---|---|---|
| 1 | People important to me think that I should resell, donate, or reuse home textiles. | 1 5 | 2 | 3 | 4 |
| 2 | Generally speaking, I want to do what my friends think I should do. | 1 5 | 2 | 3 | 4 |

*Appendix A.5. Section 5*

**Table A5.** Home Textile Disposal Intention.

| Question | Strongly disagree (1)—Disagree (2)—Neutral (3)—Agree (4)—Strongly Agree (5) | | | |
|---|---|---|---|---|
| 1 | I intend to resell my used home textiles to others directly or through a retailer. | 1 5 | 2 | 3 | 4 |
| 2 | I intend to donate my used home textiles to a charitable organization or cause. | 1 5 | 2 | 3 | 4 |
| 3 | I intend to reuse my used home textiles for other purposes. | 1 5 | 2 | 3 | 4 |
| 4 | I intend to repair my home textiles when damaged. | 1 5 | 2 | 3 | 4 |
| 5 | I intend to throw my used home textiles in the trash. | 1 5 | 2 | 3 | 4 |

*Appendix A.6. Section 6*

**Table A6.** Home Textile Disposal Behaviour.

| | | Resell | Donate | Reuse/store | Repair | Discard |
|---|---|---|---|---|---|---|
| 1 | Your bedlinens have a hole. | | | | | |
| 2 | The curtains in the living room are in good condition but you want to change them. | | | | | |
| 3 | The tablecloth has a stain that doesn't go away. | | | | | |
| 4 | The colour of the chair pads in the kitchen is faded. | | | | | |
| 5 | There are towels taking space in the cupboard that have never been used. | | | | | |
| 6 | The furniture in the bedroom has been changed and you need to get rid of the old pillow covers and blankets. | | | | | |

*Appendix A.7. Section 7: Demographic Information (almost done!)*

- Gender:

  ○ Male
  ○ Female
  ○ I prefer to not answer

- Age:

  ○ 18–25
  ○ 26–35
  ○ 36–45

- ○ 46–55
- ○ 56–65
- ○ Over 65

- Nationality: _____________
- Number of adults living in the house: _____
- Number of children living in the house: _____
- Education level (accomplished):

    - ○ Middle School.
    - ○ High School.
    - ○ Bachelor Degree.
    - ○ Master Degreed.
    - ○ PhD.
    - ○ Other: ___________

- Monthly disposable Income (SEK):

    - ○ < 5.000
    - ○ 5.000–10.000
    - ○ 10.000–20.000
    - ○ 20.000–30.000
    - ○ 30.000–40.000
    - ○ 40.000–50.000
    - ○ >50.000
    - ○ I have no personal income, but I share a household income.
    - ○ I prefer not to answer

Do you have any additional comments about this survey? Is there anything you want to share with us? You are more than welcome to provide any comment; keep in mind that we will not be able to identify or get in touch with you since the survey is anonymous.

**Appendix B.**

Interview guide for consumers

Preconditions This is something that can affect the respondent and is therefore good to understand. The preconditions are often found out if you deepen in the other areas, so these questions might not be asked by themselves (if they do not come in naturally). However, for the schedulee this can be a good way to start the conversation.

- Where are you from? Where do you live?
- What is your occupation?
- Who do you live with?
- How do you get around?

Ice-breakers:

- Do you feel that you have enough space at home? (we are looking for how they live and how they perceive their space of living … ).
- Do you see textiles as something valuable? (if they see only a functional value or also some emotional).

Recycling of textiles

- What do you do with textiles you no longer use?
- What kind of textiles do you think of?
- When do you sort textiles? When would you like to do it?

Drivers and Motivation

- What would you like to do with your used (home) textiles?
- What is stopping you?
- How do you feel after sorting/recycling (textile)?
- How would you like to feel?

Sustainable lifestyle

- What is a sustainable lifestyle for you?
- What do you think when you hear the word "sustainable" or sustainable life at home"?
- What do you throw away in your regular bin?
- What do you not sort today?
- Is there anything in your home that you do not sort?
- Is there anything that you (would) like to sort?

Consumption habits

- What is important for you when you get new things recently?
- When do you get new things?
- How often would you say that you get new things recently? (estimate)

Responsibility

- Who is responsible for taking care of textiles (that you no longer use)?
- What is your responsibility?
- Who do you think should be responsible for reducing waste in home textiles and why?
- If you knew that . . . . would you act differently?

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
