# Peer review of "Circular Economy in Home Textiles: Motivations of IKEA Consumers in Sweden"

_sustainability, doi:10.3390/su12125030_

Round 1
Reviewer 1 Report
Kindly find the attachments.

Reviewer 2 Report
Your study is timely, relevant and interesting. It is generally well-developed and well-written. However, there is a lot of room for improvement in the description of the method and results sections.
First of all, in your literature section, there are a number of issues you may want to clarify.
On p.3, please give some examples of economic incentives/motivations.
On p.4, your section on social motivations is somewhat muddled and confusing. It seems a mixture of moral beliefs, is overly focused on donation behavior. And the 'social' angle in sustainability you discuss is not about moral beliefs but about respecting a sustainable social behavior (e.g. paying decent wages, protect workers...). You may want to make this section more focused. You may also want to consider to relabel it into ‘moral beliefs’, because that may come closer to what you actually mean. If you stick to ‘social motivations’, you should stay closer to the meaning of that construct.
On p.4, you write: “A report by Morgan & Birtwistle [25] on fast fashion consumers emphasized that consumers’ lack of awareness about the textile waste impact and individual responsibility, and therefore a lack of sense of guilt, is highly influential in their disposal patterns. Similar conclusions were reached by Joung [22]. Domina & Koch [18] observed that discarding of textiles is usually provoked by the lack of awareness about alternative disposal options.” - This does not seem to have anything to do with environmental concerns. It thus not seem to belong in this section.
On p.5 you write: “There appears to be a curvilinear relationship between the family’s social class and the joint involvement of partners in the decision-making process, showing that middle-income families make more decisions together.” – This does not seem to fit in the 'normative' motivations category. Please consider removing it or justify why you think it belongs there.
From p.6 onwards, you describe your method. Part of your method is under-reprted, and I also suggest you rearrange the materials in this section. Provide separate sections on: procedure (amongst others how respondents were selected and approached, sample composition, measures, flow of the questionnaire, data cleaning… What is the link between Table 1 and the questionnaire in appendix? This is confusing: the questionnaire in appendix contains different constructs, different items etc. than the one reported in Table 1. p.7. Finally, you need to indicate the relevance of your qualitative study and the link between this qualitative part and the literature review and the quantitative study. You also need to describe the sample and the procedure, including some justification for the structure and content of the interview guide. Also explain how you analyzed these qualitative data.
In both the method and the results section, there are things that are not well explained and confusingly reported. First of all, for everything you analyze, you should clearly report which variables you used and in what way. What were the dependents and independents, how were they measured. Show, for instance, regression model structure. More specifically, I am confused and worried about how you used your independents (disposal behavior). Although you do not report this in the text, I infer from the appendix that they are measured in terms of ‘yes’ or ‘no’: in the past 12 months, either people did something or they didn’t. Hence, you are able to report percentages, as you did in the figure. However, you also used these dichotomous variables as independents in your regression analysis. This seems to be statistically incorrect. When a dependent is dichotomous, logistic regression should be used. Please do, or otherwise justify why you used ordinary multiple regression analysis. Also, to make your point, you do not have to report the correlational analysis, just the regression analyses are enough. In reporting your regression results, you should mention which variables have a positive and negative effect, and which effects are stronger or weaker than others. In order to do the latter, you need to report and use Beta coefficients.
Some other issues:
p.8: ‘estimated values’ - which values do you mean?
p.9: “when XER was held constant”: what do you mean?
p.9. Report p<.001 instead of p=.0000
p.9. what is ‘charity’ doing here?
p.9. “as expected”. Why expected? You did not formulate expectations or hypotheses.
p.13: Spilt your ‘conclusion’ section up in two parts: ‘conclusions’ and 'managerial/public policy implications'.
Reviewer 3 Report
Dear Authors,
the topic is very interesting and new (the exploration of consumer behaviour and reasons for home textile consumption and disposal practices).
However the paper appears quite confused starting from the abstract until the whole document, it seems more a master thesis than a scientific academic paper.
For instance, there is a great confusion about the methodology: is it a case study on IKEA’s 97 customers (on page 3, line 93 “IKEA’s 97 customers in Sweden as case study”) or on IKEA (see on page 5, line 206: We used ikea as case study)? Or is it a quantitative analysis base on a regression? Moreover, with reference to qualitative data collected by semi-structured interviews with 24 Swedish consumers, the authors did not collect directly them but, on the contrary, they had access to qualitative interviews conducted by IKEA staff.
Furthermore, the most part of the paper describes the carried out regression analysis instead of discussing the results in light of the theoretical background.
The contribution of the research is not well expressed and theoretical and managerial implications are missing.
Round 2
Reviewer 2 Report
You have done a good revision job and the manuscript has improved substantially. However, there are still some issues to be resolved.
On p.4, in one and the same paragraph, you talk about ‘social norms and conventions’ and ‘moral beliefs’. Further in the same paragraph, you talk about social learning, and also about prosocial motivations. You give the impression that these are all more or less the same. I already pointed at this in my first review. Since you don’t do anything with ‘moral beliefs’, you may want to avoid using it in your argumentation. Also ‘prosocial motivations’ are not the same as ‘social norms and conventions’. The latter come close to the ‘subjective norm’ factor. Please clean up this conceptually muddled section and develop your arguments here such that they reflect the variables you are going to eventually measure and use in your empirical study.
On p.6 and further, there is a section entitled ‘logistic regression analysis’. This section contains a lot of materials that do not belong under such a title, but should be reorganized in different sections. On p.8 in this section, you all of a sudden formulate 5 hypotheses that fall out of the blue sky. First of all, hypotheses development should come right after the literature review (or be developed using the literature), and BEFORE the method section. Second, you did not develop these hypotheses at all: you do not argue why some motivations and not others would be relevant for some types of disposal, but not for others. It is essential that you justify this, based on your literature review and deductive logic. Second, the first part of the section should go under the heading ‘measures’. In this section you should explain which variables you measured and how. As I already mentioned in my previous review, there is no consistency between what you report in the text and the questionnaire in the appendix. The questionnaire seems to contain many more items than you used to define your variables for the empirical analysis. This is very confusing. Please clean up and justify why you used which items. Also, your variable ‘normative issues’ is rather vague: what exactly do you mean? The three items same to measure different things, for instance NI1 seems to refer to ‘subjective norm’ and NI3 to ‘convenience’. I am not surprised that alpha is low for this variable. Please define and measure this variable better and rerun the analyses. On p.7, you do not justify why you finally selected the items you did and left out others.
From p.8 onwards, you should add a title ‘analyses’. Why did you use multinomial logistic regression and not simply multiple regression? After all, you have measured everything on 5-point scales. On p.10, your table 3 is unclear, and not sufficiently explained. For instance, in the title there is ‘likelihood ratio (chi²)’, but this is not in the table. Also: clarify what the numbers in the cells of the table represent. Please also discuss the relative importance of motivations based on the results. For instance, with ordinary multiple regression you could do that by inspecting the Betas. I am sure you can also do that in logistic regression.
Author Response
Thank you again for your invaluable feedback. We hope that our work to revise the manuscript along your suggestions improved it. Find attached our specific replies to your comments.

Reviewer 3 Report
Dear Authors,
I found the paper improved. In its current form, it is clearer and has a more scientific structure.
It is ready for publication!
Best regards
Author Response
We want to thank reviewer 3 for the contributions to the improvement of our manuscript, and we are glad that our improvements satisfy reviewer 3.
Round 3
Reviewer 2 Report
No more comments or suggestions